# Identification of Driver Epistatic Gene Pairs Combining Germline and Somatic Mutations in Cancer

**DOI:** 10.3390/ijms24119323

**Published:** 2023-05-26

**Authors:** Jairo Rocha, Jaume Sastre, Emilia Amengual-Cladera, Jessica Hernandez-Rodriguez, Victor Asensio-Landa, Damià Heine-Suñer, Emidio Capriotti

**Affiliations:** 1Department of Mathematics and Computer Science, University of the Balearic Islands, 07122 Palma de Majorca, Spain; 2Genomics of Health Group, Health Research Institute of the Balearic Islands (IDISBA), 07120 Palma de Majorca, Spain; 3BioFolD Unit, Department of Pharmacy and Biotechnology (FaBiT), University of Bologna, 40126 Bologna, Italy

**Keywords:** gene pairs, epistasis, cancer driver variations, contingency table, survival analysis, lung cancer, colon cancer

## Abstract

Cancer arises from the complex interplay of various factors. Traditionally, the identification of driver genes focuses primarily on the analysis of somatic mutations. We describe a new method for the detection of driver gene pairs based on an epistasis analysis that considers both germline and somatic variations. Specifically, the identification of significantly mutated gene pairs entails the calculation of a contingency table, wherein one of the co-mutated genes can exhibit a germline variant. By adopting this approach, it is possible to select gene pairs in which the individual genes do not exhibit significant associations with cancer. Finally, a survival analysis is used to select clinically relevant gene pairs. To test the efficacy of the new algorithm, we analyzed the colon adenocarcinoma (COAD) and lung adenocarcinoma (LUAD) samples available at The Cancer Genome Atlas (TCGA). In the analysis of the COAD and LUAD samples, we identify epistatic gene pairs significantly mutated in tumor tissue with respect to normal tissue. We believe that further analysis of the gene pairs detected by our method will unveil new biological insights, enhancing a better description of the cancer mechanism.

## 1. Introduction

Cancer is a complex disease driven by several factors [1,2]. No single genetic factor alone can explain cancer onset; thus, a multigenic mechanism should be taken into consideration [3,4]. In this context, epistasis analysis allows the identification of gene-pair interactions associated with the insurgence and progress of cancer.

One of the main challenges in current genomics is to identify the correlation between genotype and phenotype. GWAS (genome-wide association analysis) experiments have attempted to establish this association between genotype and phenotype by relying on SNP (single-nucleotide polymorphism) data. However, there is only a small proportion of the phenotype that can be explained with common SNPs, raising the well-known debate about “missing heritability” [5].

Usually, SNPs are tested for their statistical relationship with a disease considering only additive effects. In other words, each SNP is presumed to independently contribute to the phenotype. Nevertheless, it has been shown that missing heritability may arise from genetic variants that exhibit effects through interactions with one another [6,7]. Under this condition, the  concept of epistasis becomes relevant, referring to the combinatorial effects of one or more genetic variants that can manifest even without any individual effect. We believe that complex traits would be better explained by considering the interaction between pairs of genes and variants. Our work aims to uncover novel gene pairs associated with the onset and progression of cancer.

Research consortiums such as The Cancer Genome Atlas (TCGA) [8] and the International Cancer Genome Consortium (ICGC) [9] have generated a huge amount of cancer genome data, which has allowed the discover y of potential causative variants [10]. Different authors have reviewed the computational methods available for assessing the impact of mutations on the cancer genome [11,12,13].

Several methods have been used to discriminate between driver mutations, which provide a selective advantage to the cancerous cells and contribute to the onset of the disease [14,15,16,17], and passenger mutations, which have no pathogenic effect. The prevalent strategy to identify cancer driver genes works by detecting significantly over-mutated genes in tumors, which are more likely the drivers.

One potential strategy for identifying gene interactions can involve conducting exhaustive tests on all combinations of variants. Many software packages consider statistical interactions between loci due to this complex mechanism that involves multiple genes. For example, Marchini’s algorithm and PLINK [18,19] test for all two-locus interactions in a reasonable timeframe. PLINK has a specific option (--epistatic) to discover such pairs. However, the  TCGA datasets consist of two samples per subject, encompassing both normal and tumor tissues. Consequently, the experimental design based on paired data from the same subject challenges the hypothesis of the sample independence assumed by regression methods.

Similar to PLINK, BOOST [20] uses general linear regression models over all possible gene pairs. However, it adopts a fast approximation strategy that guarantees that significant interactions are not filtered out. Nevertheless, interpreting the regression differences is not straightforward. This is because BOOST is specifically designed for the analysis of case–control data, which is not directly applicable to our problem that involves paired data.

An alternative approach is MOSGWA [21], which addresses the selection of interacting genes as a variable selection problem using a modified version of the Bayesian information criterion (mBIC2). When compared with methods employing a similar strategy [22,23], MOSGWA resulted in a lower fraction of false positives.

A specific study on colorectal cancer [24] presented a simple but powerful method for testing gene–gene interactions, reducing the number of pairs by considering only the variants with a certain marginal association with cancer. The authors discovered two significant interactions involving known loci and variants that exhibited marginal associations.

Other approaches by Vandin and colleagues [25,26] focused on the identification of cancer-associated gene networks. GeneralizedHotNet [25] detects sub-networks of mutated genes within established or known interaction networks, correlating them with a given phenotype. The procedure is based on a heat diffusion process to obtain a measure of influence between pairs of genes in a protein–protein interaction (PPI) network. A two-stage statistical test scoring the association between mutated genes and clinical data is adopted for the selection of significant subnetworks. A more recent method, namely NoMAS [26], identifies sub-networks of a large gene–gene interaction network with mutations associated with survival time. NoMAS implements an efficient algorithm that leverages a color-coding technique and a log-rank statistical test to compare the survival of two specific populations.

In this study, we present a new method for the identification of gene pairs associated with cancer by comparing samples of normal and tumor tissues. Unlike previous methods developed by Vandin and colleagues [25,26], which rely on known protein–protein interaction (PPI) networks, our approach does not assume any predefined gene pairs. Our algorithm represents paired data in a contingency table and, focusing on specific elements of the table, assesses the statistical significance of cases of mutated gene pairs, where at least one of the genes exhibits a somatic variant. To the best of our knowledge, our method is the first to consider the association between somatic and germline mutations. Furthermore, the analysis proposed in this study enables the identification of cancer-causing gene pairs, wherein each single gene by itself does not exhibit a significant association with cancer.

## 2. Results

Analyzing the variant calling files, which include the detected variants in normal and tumor tissues, we calculated a 3 × 3 contingency table to identify patients holding relevant gene-pair mutations (RGPMs) corresponding to the states (s,s), (s,b), and (b,s). Indeed, we assume that epistasis effects in cancer can not only be attributed to pairs of somatic (*s*) variations but can also arise from the combination of a germline (*b*) and a somatic (*s*) variant within a given gene pair. This condition is equivalent to a congenital predisposition associated with a germline variant, wherein the interaction with a somatic variant confers a selective advantage to the cell.

To identify relevant epistatic interactions, we employed two criteria when evaluating all potential gene pairs. The first criterion involved selecting gene pairs that exhibited a significant number of patients with an RGPM in the contingency table. The second criterion entailed identifying gene pairs with a significant difference in overall survival (OS) time between subjects with RGPMs and those with a background single gene mutation (BSGM) corresponding to the states (w,b), (w,s), (b,w), and (s,w). The definition of gene pair mutation states and the procedure for calculating the contingency table are described in the Materials and Methods section.

Despite considering all gene-pair candidates, the procedure itself is remarkably fast. The output of the analysis is a comprehensive list of gene pairs along with their corresponding *p*-values, which show significant associations with cancer after correcting for the multiple testing hypothesis.

### 2.1. Colon Cancer

#### 2.1.1. Detection of Epistatic Interactions

We applied our method to a TCGA dataset comprising 422 patients affected by COAD. Our analysis detected 358,774 variants affecting a protein sequence (VAPs) that appear in at least one individual. After the filtering procedure, 11,670 genes were retained, and ∼68 million (11,670 × 11,669/2 = 68,088,615) gene-pair contingency table were calculated for further analysis. Among them, we selected ∼51 million (51,098,733) gene pairs for which the number of observed patients with any RGPM was higher than the relative the expected value. To keep the false positive rate below 5%, gene pairs with a *p*-value lower than $9.8\times 10^{-10}$ (0.05/51,098,733) were selected. The results of analysis of COAD samples are summarized in Appendix A.

After this filtering procedure, 450 gene pairs passed the test (Appendix A). We performed a survival analysis with the Bonferroni correction on this subset, searching for the gene pairs with an associated *p*-value lower than $1.1\times 10^{-4}$ (0.05/450).

Although no pairs pass this test, we found 16 gene pairs with a *p*-value under 0.05. For each selected gene pair, Table 1 presents the *p*-values and the hazard rate (HR), which measures the relative rate of deaths comparing two groups of patients. On average, for the 16 gene pairs, the group of individuals with RGPMs doubles the death rate of the group of patients with BSGMs.

Furthermore, we studied the association between the two groups of patients and the clinical data provided by the TCGA consortium. In detail, we conducted regression analyses to detect the potential associations between tumor stage, gender, and age at initial diagnosis in relation to patient class. However, no significant associations were found in these analyses (*p*-values not shown).

All contingency tables were analysed in less than 3 min at 2.6 GHz with a program written in C [27]. The survival analysis of the pairs that pass the first test was performed in less than a minute with a dedicated Python package [28].

To further characterize the type of diseases associated with the genes that are selected through the epistasis test, we performed an enrichment analysis of the genes within the selected pairs using EnrichR [29]. In  Figure 1A, we displayed the Jensen’s disease entities [30] with the highest level of significance associated with the list of selected genes. It is evident that the most significant entities are related to different types of carcinoma. This observation underscores the effectiveness of our selection procedure in capturing crucial biological aspects associated with the mechanisms of cancer. The robustness of our procedure is demonstrated by the consistent results obtained even when the larger list of the top 1000 gene pairs is considered (Figure 1B).

Finally, we analyzed the contingency tables of the first two significant gene pairs (Table 2). Although these gene pairs do not meet the criteria for significance according to the Bonferroni test, their survival analysis *p*-value is lower than 0.05.

For the gene pair CCDC73-HTR2B (Table 2A), the number of observed subjects for the entry (s,s), corresponding to the case where both genes hold somatic mutations, is 8, which is significantly higher than the expected value of 0.36. As shown in Figure 2A, the average overall survival time of the 10 patients with relevant gene-pair mutations (RGPM) is 212 days shorter than the value calculated for the subjects with a background single gene mutation (BSGM). For this gene pair, the  survival analysis *p*-value is 0.03. Similarly, for the gene pair DDX4-KCNJ16 (Table 2B), the number of subjects observed for the entry (s,s) is 6, which is significantly above the expected value of 0.17. For the group of 8 patients holding a RGPM, the average overall survival time is 355 days shorter than the value calculated on the set of individuals with a BSGM (Figure 2B). For this gene pair, the survival analysis *p*-value is 0.05. Figure 3 shows the VAPs that passed the filters and generate the mutations in the four genes.

We aim to discover pairs of loci that are not known to act together in the tumor cells. However, some evidence may already be present elsewhere. We will observe some relationships with gene function in public databases and with protein interaction networks.

#### 2.1.2. Gene and Protein-Knowledge-Based Analysis

To generate new hypotheses regarding the mechanism of cancer, we performed an analysis of the functions of the identified genes and their corresponding protein interaction networks. A previous study has shown that lung cancer patients with a high transcription level of the CCDC73 gene tend to have a good prognosis [32]. Another study found a relationship between the expression of the serotonin 2B (HTR2B) receptor and human uveal melanoma [33]. Nevertheless, no relationship with colon cancer has been found until now. Based on previous studies, DDX4 has already been recognized as a potential molecular target for chemotherapy due to its involvement in regulating cell cycle progression in various somatic-derived blood cancer cells [34]. Furthermore, KCNJ16 is one of the 154 signature genes that could be used to differentiate carcinoma types [35] although no direct association with colon cancer has been reported to date. Therefore, our approach was able to identify two potential candidate gene pairs that could be proposed as molecular targets for the development of colon cancer therapies.

In the database of known and predicted protein–protein interaction STRING [36], neither the genes CCDC73 and HTR2B nor DDX4 and KCNJ16 interact, not even at a low confidence level. In Wikipathways [37], HTR2B, DDX4, and KCNJ16 do not share pathways, while CCDC73 is not present. HTR2B, DDX4, and KCNJ16 do not share pathways.

Therefore, the interactions between the gene pairs identified in this work are currently unknown and require further investigation.

### 2.2. Lung Cancer

#### 2.2.1. Detection of Epistatic Interactions

As a second case study, we applied our approach to the analysis of a dataset of 405 subjects with lung adenocarcinoma (LUAD), identifying 192,304 VAPs. After performing a filtering procedure, 6854 mutated genes were retained for further analysis. Considering these 6854 mutated genes, we calculated a contingency table for ∼23 million (6854 × 6853/2 = 23,485,231) gene pairs. Among them, 10,961,208 had at least one RGPM with a number of observations above the expected number. To keep a false positive rate below 0.05, a  *p*-value threshold of $5\times 10^{-9}$ (0.05/10,961,208) was considered. The results of analysis of LUAD samples are summarized in Appendix A.

With this *p*-value cut-off, only four gene pairs pass the filter, but none of them pass the survival difference significance test. Given the limited number of significant gene pairs, we relax the *p*-value cut-off to recover other possible pairs. We consider the top 100 pairs (Appendix A).

The *p*-value of the differences in the overall survival time was calculated for each of the top 100 pairs (epistasis *p*-value $<7.32\times 10^{-7}$). With a *p*-value threshold of 5.0 × $10^{-4}$ (0.05/100) corrected for the Bonferroni multiple testing hypothesis, no gene pairs passed this second control. Although no pairs passed this test, we found three gene pairs with a *p*-value under 0.05 (Table 3). For each selected gene pair, Table 3 presents the *p*-values and the hazard rate (HR), which measures the relative rate of deaths comparing two groups of patients. Similar to the observations in COAD, in the case of LUAD, the group of individuals with RGPMs, on average, is double the death rate of the other group of patients.

To characterize the type of diseases associated with the selected gene pairs, we performed an enrichment analysis using EnrichR [29]. In Figure 4A, we displayed the Jensen’s disease entities [30] with the highest level of significance associated with the list of selected genes. Additionally, in the case of lung cancer, the most significant entities are related to different types of carcinoma. The robustness of our algorithm is demonstrated by the consistent results obtained even when the larger list of the top 1000 gene pairs is considered (Figure 4B).

Let us examine the gene pairs CHRM2-SLC6A15 (Table 4A) and PSD2-SUGP1 (Table 4B), which are the top epistatic pairs that pass the survival significance threshold.

The number of observed subjects for the entry (s,s) of the gene pair CHRM2-SLC6A15 is 5, which is significantly above the expected value of 0.19. For this gene pair, the epistatic *p*-value is $1.2\times 10^{-8}$. Analysing the contingency table of the gene pair PSD2-SUGP1, we found that for (s,s), the observed and expected subjects are 3 and 0.08, respectively, while for (s,b), the observed and expected subjects are 4 and 1.72, respectively. The epistatic *p*-value for the PSD2-SUGP1 gene pair is $3.52\times 10^{-7}$.

The survival analysis for the CHRM2-SLC6A15 gene pair shows that the subjects with RGPMs have 137 fewer days of survival on average than the subjects with BSGMs (Figure 5A). This difference in the overall survival time corresponds to a *p*-value of 0.042. For the PSD2–SUGP1 gene pair, the average overall survival time difference between the patients with RGPMs and those with BSGM is 421 days (Figure 5B), which corresponds to a *p*-value of 0.006. Although these *p*-values are lower than 0.05, they do not pass the control for the multiple testing hypothesis (q-value $<5.0\times 10^{-4}$).

Figure 6 shows the VAP positions that passed the filters and generate the mutation of the pair of genes CHRM2-SLC6A15) and PSD2-SUGP1.

#### 2.2.2. Gene and Protein-Knowledge-Based Analysis

To gain new insight into the mechanism of lung adenocarcinoma, we analyzed the existing information on the four genes forming the top two gene pairs. CHRM2 has been associated with nicotine dependence [38], while SLC6A15 has been associated with depression disorders [39]. Furthermore, SUGP1 has been recently linked to lung cancer [40], and PSD2 has been linked to neurological diseases according to GeneCards. These findings constitute an initial validation of the association between the selected gene pairs and lung cancer, providing early evidence to support their potential significance.

In STRING, none of the pairs have an interaction, not even at the low confidence level, nor a common interacting protein at a low confidence level. Looking in Wikipathways, the two gene pairs have no pathways in common. Thus, the cancer interactions between the two top gene pairs detected with our approach are not yet known.

## 3. Discussion

This research proposes a new method to identify gene pairs potentially associated with cancer. In particular, our approach is based on the analysis of paired normal and tumor samples from the same patient and excludes already known driver genes. The procedure developed to detect epistatic gene pairs is effective and fast, as the analysis of the contingency tables that we defined is rigorous and straightforward. The new statistical framework, based on the analysis of specific table cells, correctly characterizes the epistatic gene pairs and can be used in case of dependencies among paired samples from normal and tumor tissues.

In detail, we present an analysis of four epistatic gene pairs detected in colon and lung cancer. Although no protein–protein interactions between these gene pairs have been identified, the results of the survival and the disease enrichment analysis are promising. The enrichment analysis of diseases suggests that the process presented allows the detection of genes associated with cancer, and  candidate epistaic pairs can be validated combining these results with clinical data.

In conclusion, the identification of epistatic gene pairs enables the formulation of new hypotheses on the mechanisms of cancer that should be validated through targeted experimental assays.

## 4. Materials and Methods

### 4.1. Statistical Analysis of Genomic Association

A statistical analysis designed to detect significantly mutated gene pairs relies on the calculation of a contingency table. Given a gene *g* in a patient, it can assume three possible states:*w*: the gene is mutated neither in normal and tumor tissues;*s*: the gene is mutated only in the tumor tissue (somatic mutation);*b*: the gene is mutated in both normal and tumor tissues (germline mutation).

In this study, we focus on the patients holding the following relevant gene-pair mutations (RGPMs):
(s,s): both genes present somatic mutation, being mutated only in tumor tissue;(s,b), (b,s): one gene carries a germline mutation (both in normal and tumor tissues), and the second gene presents a somatic variant (only in tumor tissue).

Considering the three possible gene states i,j∈{w,s,b}, the following notations can be adopted for calculating the contingency table:$o_{ij}$ is the number of observed subjects in the cell (i,j);$e_{ij}$ is the expected number of subjects in the cell (i,j);$m_{i}$ is the row marginal frequency;$n_{j}$ is the column marginal frequency, and *N* is the total number of subjects.

For each pair of genes $g_{1}$ and $g_{2}$, we focus on the elements of the contingency table accounting for the three case studies mentioned above:    
$g_{1}/g_{2}$wsb*w*    

*s*
$o_{ss}$$o_{sb}$*b*
$o_{bs}$


    To identify potential gene interactions, an independence test on the contingency table could be conducted. However, our specific interest lies in testing the dependency exhibited by the three pre-defined elements. For this purpose, we performed a test by considering contingency tables with identical marginal frequencies while fixing the three cells, aiming to find gene pairs with minimum dependency.

Given that the number of degrees of freedom is 4, fixing the value of the three cells, only one degree of freedom is left. Consequently, our strategy consists of finding the optimal cell value that minimizes the G-test statistic. This is a minimization problem over a single variable subject to the non-negativity constraints of all the table cells.

If we define *x* as the new value of $o_{bb}$, then the other five cell values can be calculated as a function of *x*, and the statistic of the following table can be a function of *x*:   
$g_{1}/g_{2}$wsb*w*$c_{3}+x$$o_{ws}$$c_{1}-x$*s*$o_{sw}$$o_{ss}$$o_{sb}$*b*$c_{2}-x$$o_{bs}$*x*where $c_{1}=n_{b}-o_{sb}$, $c_{2}=m_{b}-o_{bs}$ and $c_{3}=n_{w}-m_{s}+o_{ss}+o_{sb}-m_{b}+o_{bs}$ are the constant values fixed by the marginal constraints.

Notice that under the null hypothesis of independence, the values $e_{ij}$ are defined only on the marginals and therefore do not depend on *x*. Additionally, the values of $o_{sw}$ and $o_{ws}$ are constant when the other two cells in the same row or column are constant.

The cells must be positive or zero and satisfy the following positivity constraints:$$\begin{array}{cc} o_{bb}\left(x\right)= & x\geq \;0\\ o_{wb}\left(x\right)= & c_{1}-x\geq \;0\\ o_{bw}\left(x\right)= & c_{2}-x\geq \;0\\ o_{ww}\left(x\right)= & c_{3}+x\geq 0, \end{array}$$

The *G* statistic can be defined as follows:$$G\left(x\right)=2\sum_{ij}o_{ij}\left(x\right)\ln\frac{o_{ij}\left(x\right)}{e_{ij}}$$

After some simplifications, its differential can be expressed as:$$G^{\prime }\left(x\right)=2\ln\frac{(c_{1}+x)x}{\left(c_{2}-x\right)\left(c_{3}-x\right)}.$$

Thus, $G^{\prime }\left(x\right)$ is null when
$$x^{*}=\frac{c_{1}c_{2}}{c_{1}+c_{2}+c_{3}}.$$

The second differential has four terms, all of which are positive under the positivity constraints. Consequently, the function G(x) is strictly convex, and the $x^{*}$ value corresponds to a minimum.

The denominator does not present any singularities because
$$\begin{array}{cc} c_{3} & =-o_{sw}+n_{w}-m_{b}+o_{bs}\\ & =-o_{sw}+n_{w}-o_{bb}-o_{bw}\\ & =o_{ww}-o_{bb}, \end{array}$$
and
$$\begin{array}{cc} c_{1}+c_{2}+c_{3} & =n_{b}-o_{sb}+m_{b}-o_{bs}+o_{ww}-o_{bb}\\ & =o_{wb}+m_{b}-o_{bs}+o_{ww}\\ & =o_{wb}+o_{bw}+o_{bb}+o_{ww}, \end{array}$$

If the denominator in the equation for $x^{*}$ is 0, all the four corner of the table are 0, and the remaining table values are fixed. In such cases, we set $x^{*}=o_{bb}=0$.

The positivity constraints are satisfied by $x^{*}$, which is obtained by a non-negative numerator and denominator. Furthermore, $x^{*}\leq c_{1}$ since $c_{1}+c_{3}=o_{wb}+o_{ww}\geq 0$, and symmetrically, $x^{*}\leq c_{2}$.

For $c_{3}$, the following conditions are valid. If $c_{3}\geq 0$, then $x^{*}\geq -c_{3}$.

If $c_{3}<0$, then $x^{*}\geq -c_{3}$, but in the latter case, the proof is more technical. First, we observe that $o_{bb}=o_{ww}-c_{3}$; thus, $c_{1}=o_{wb}+o_{bb}=o_{wb}+o_{ww}-c_{3}$, and $c_{1}\geq -c_{3}$. Similarly, $c_{2}\geq -c_{3}$. If we fix $c_{3}<0$ and define the function
$$f\left(c_{1},c_{2}\right)=\frac{c_{1}c_{2}}{c_{1}+c_{2}+c_{3}},$$
then
$$\nabla f=\left(\frac{c_{2}\left(c_{2}+c_{3}\right)}{{\left(c_{1}+c_{2}+c_{3}\right)}^{2}},\frac{c_{1}\left(c_{1}+c_{3}\right)}{{\left(c_{1}+c_{2}+c_{3}\right)}^{2}}\right).$$

The gradient is null when $c_{1}=c_{2}=-c_{3}$, and it has only non-negative components in the quadrant $c_{1},c_{2}\geq -c_{3}$. Therefore, *f* has a minimum at $\left(-c_{3},-c_{3}\right)$ with a value of $f\left(-c_{3},-c_{3}\right)=-c_{3}$ and 
$$f\left(c_{1},c_{2}\right)\geq -c_{3}.$$

In summary, we have a value $x^{*}$ that generates a new valid contingency table with a minimum $G(x^{*})$ for any table. We take this value as a measure of dependency of the three relevant cells we have focused on. To calculate the *p*-value, we generate random permutations of subjects for one of the genes and calculate the values of the minimum statistic for ∼10 billion tables. The whole process ∼10 h on a 2.6 GHz computer with 16 processors.

The mutation dependency of two genes can be related to observing more or fewer subjects than expected in a cell. We assume that tumor is caused by the accumulation of mutations. Thus, we retain statistically significant gene pairs where at least one of the three cells has more observations than expected.

We select the gene pairs that pass the Bonferroni correction with the FWER (family-wise error rate) *Q* of 0.05 and a total number of experiments equal to the number of considered contingency tables.

### 4.2. Survival Analysis

We analysed the survival data available at the TCGA to select possible clinically relevant epistatic gene pairs. We used overall survival (OS) time and not the disease-free interval (DFI) time because the latter is related to the cancer recurrence; therefore, it would be more appropriate for a study on treatment effectiveness. Since our study refers to the overall malignity of the mutations, we considered the OS time.

For each gene pair, we categorized the patients into two groups based on their gene mutation state (w,s,b) and compared their OS times. The first group consisted of patients holding one of the three RGPMs (s,b), (b,s) and (s,s). The second group comprised individuals who had either a germline or somatic variant in only one gene of the pair. The mutation states of these individuals, corresponding to the elements (w,b), (w,s), (b,w), and (s,w) of the contingency table, are referred to as background single-gene mutations (BSGMs).

To estimate the survival function, we used the Kaplan–Meier estimate and the Cox hazard model available in the lifelines [28] package for Python.

Of the gene pairs that showed significant gene associations, we selected the ones that showed a significant difference in the survival expectancy between double-mutated and non-double-mutated, with a Bonferroni FWER q-value of 0.05.

For each gene pair, we also considered the two subject groups defined above to check for dependencies with clinical variables, such as pathological tumor stage, gender, and age at initial pathological diagnosis. We used logistic regression to find associations between the group and the clinical variables. There were 10 levels for the tumor stage that were modelled numerically; the “discrepancy” labels were also changed to “Not Available”.

#### Algorithm Overview

The algorithm can be summarized as follows Algorithm 1:
**Algorithm 1:** Cancer Epistatic Genes Finder (CEG-Finder).**For each** pair of genes:   Calculate the 3×3 contingency table for the s,b,n values   **If**
$o_{ij}>e_{ij}$ at some of the three cells {(s,s),(b,s),(s,b)}: 2.1.   Calculate $x^{*}$ and the new contingency table2.2.   Perform the G-test for the new table and calculate the *p*-value   Select the pairs that pass the Bonferroni FWER control   Apply the survival analysis and select gene pairs with significantly lower OS time

### 4.3. Dataset Composition and Processing

We collected two datasets, with one consisting of samples of colon adenocarcinoma (COAD) and the other composed of samples of lung adenocarcinoma (LUAD), both of which were released by the TCGA consortium. For each patient, only one pair of samples was considered. The two datasets consist of N=422 and N=405 unique paired samples (normal/tumor) for COAD and LUAD, respectively. The VCF files associated with each sample were obtained considering GRCh37 and GRCh38 reference human genomes [41] for COAD and LUAD, respectively. The annotation of mutations was performed using Annovar [42].

We collected the variants affecting a protein sequence (VAPs) in each sample: non-synonymous single-nucleotide variants, frameshift deletions, frameshift insertions, stop-gain, stop-loss, non-frameshift deletions and non-frameshift insertions. We assumed that any VAP can impair gene function.

The considered VAPs were either labelled as “PASS” in the “Filter” column of the VCF or met the specific criteria for the read depth (DP) in support of the alternative allele. In detail, we filtered out the VAPs for which the DP was lower than 10 reads and for which there were a fraction of reads supporting the alternative allele (alternative alleles supporting read divided by read depth) lower than 10%. Additionally, we excluded from the analysis the VAPs with an excess of mutated cases in tumor tissue with respect to normal because they usually correspond to sequencing artefacts. To reduce the impact of these artefacts, we focused only on somatic mutations that exceed the number of cases with a germline mutation by three subjects or fewer.

A gene is considered mutated with respect to the reference genome if it contains any type of mutation at any locus. To reduce the number of genes tested, we filtered out all the genes with less than 5% of the mutated subjects in a tumor tissue.

In addition, all pseudogenes, all genes associated with olfactory receptors, and the macro-gene TTN (titin) were eliminated.

To identify pairs of genes that are not individually associated with cancer, we excluded from our analysis all the driver genes for lung or colon adenocarcinoma reported by Intogen [43].

Among the 422 patients affected by COAD, the OS time was available for 403 of them, while the information on 93 individuals was censored. In the case of the LUAD cohort, which comprises 405 subjects, the OS time is available for 392 patients, while data for 115 individuals are censored.

## Figures and Tables

**Figure 1 ijms-24-09323-f001:**
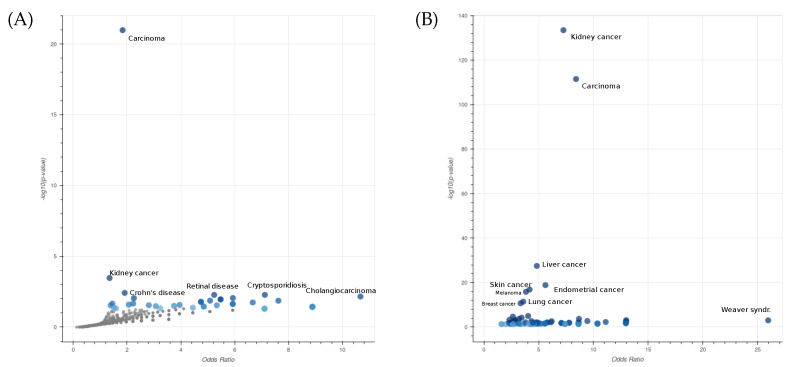
Enrichment analysis of Jensen’s disease entities performed on a list of 444 genes involved in 450 significant epistatic interactions (**A**) and a second list of 782 genes involved in the top 1000 epistatic interactions (**B**) in COAD. The x- and y-axes are the odds ratios and *p*-values of the disease enrichment, respectively. The entity Carcinoma describes a family of diseases characterized by abnormal proliferation of epithelial cells.

**Figure 2 ijms-24-09323-f002:**
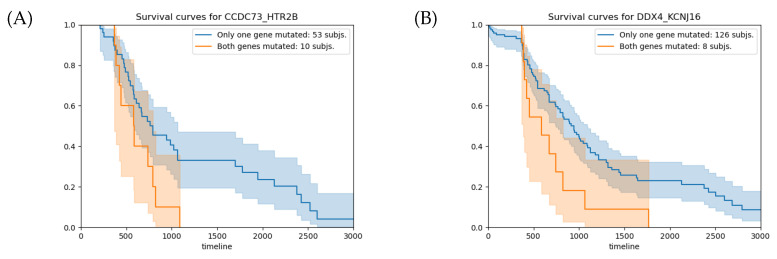
(**A**) Survival analysis for the CCDC73-HTR2B gene pair. The average overall survival times for the groups with RGPMs and BSGMs are 791 and 579 days, respectively, with a relative *p*-value of 0.03. (**B**) Survival analysis for the DDX4-KCNJ16 gene pair. The average overall survival times for the groups with RGPMs and BSGMs are 943 and 588 days, respectively, with a relative *p*-value of 0.02. The orange and blue curves represent the groups of subjects with RGPMs and BSGMs, respectively. RPGMs: (b,s), (s,b) and (s,s). BSGMs: (w,b), (w,s), (b,w) and (s,w). The mutation states of each gene are: *w* (no mutation), *s* (somatic), and *b* (germline).

**Figure 3 ijms-24-09323-f003:**
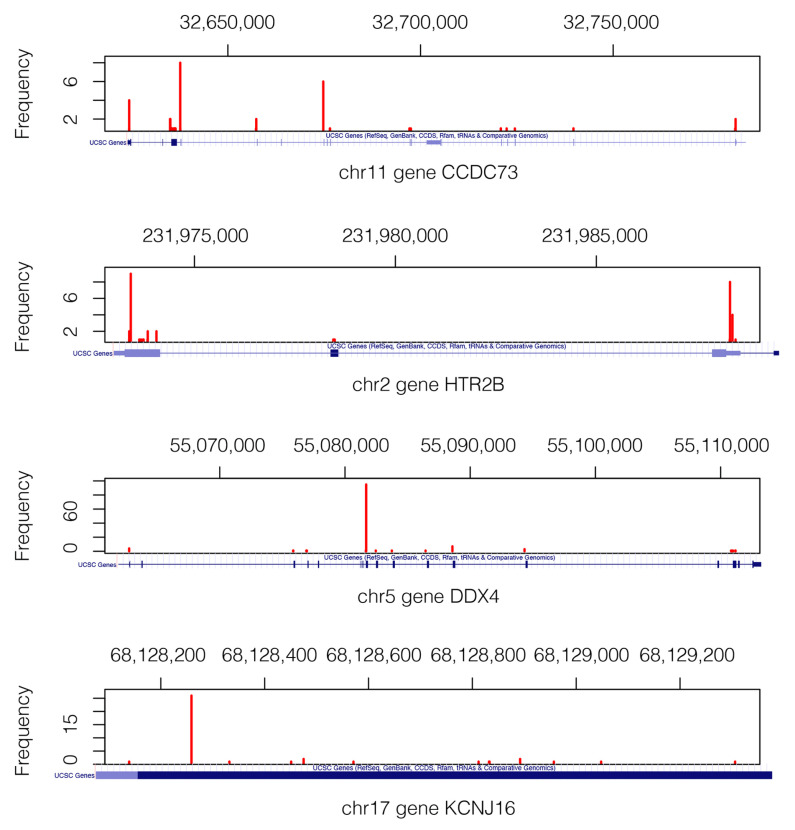
Mutation landscape in genes CCDC73, HTR2B, DDX4, and KCNJ16. Only the VAPs that pass the filters are shown. We can see where the actual VAP positions in each gene are. The gene structures are shown below each figure in a blue axis [31]. Mutations of gene KCNJ16 are in the short range of 1.2 Kb, which explains why the depicted gene structure corresponds exclusively to the exonic region.

**Figure 4 ijms-24-09323-f004:**
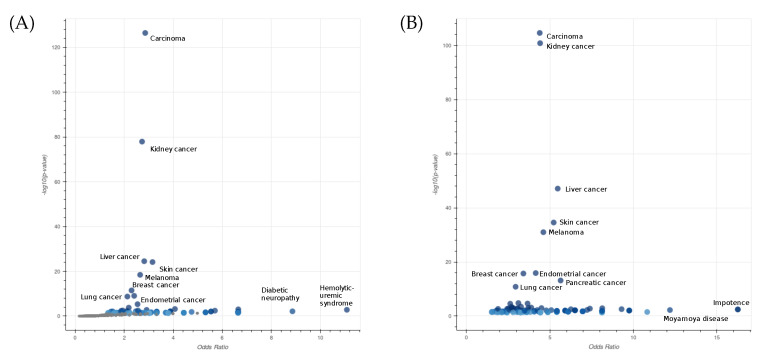
Enrichment analysis of Jensen’s disease entities performed on a list of 115 genes involved in 61 interactions with a survival significance under 0.05 among the top 1000 epistatic interactions (**A**), and a second list of 1220 genes involved in all top 1000 epistatic interactions (**B**) in LUAD. The axes x and y are the odds ratios and *p*-values of the disease enrichment, respectively. The entity Carcinoma describes a family of diseases characterized by abnormal proliferation of epithelial cells.

**Figure 5 ijms-24-09323-f005:**
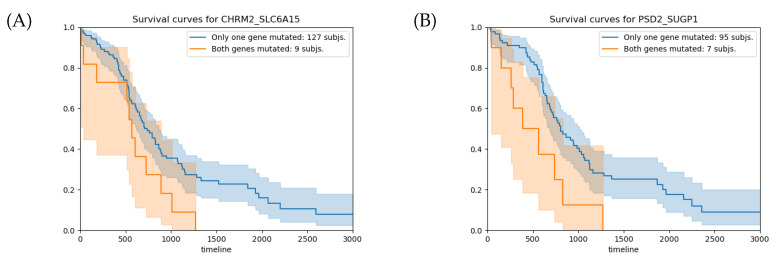
(**A**) Survival analysis for the CHRM2–SLC6A15 gene pair. The average overall survival times for the groups with RGPMs and BSGMs are 705 and 568 days, respectively, with a relative *p*-value of 0.042. (**B**) Survival analysis for the PSD2–SUGP1 gene pair. The average overall survival times for the groups with RGPMs and BSGMs are 806 and 385 days, respectively, with a relative *p*-value of 0.006. The orange and blue curves represent the groups of subjects with RGPMs and BSGMs, respectively. RPGMs: (b,s), (s,b), and (s,s). BSGMs: (w,b), (w,s), (b,w), and (s,w). The mutation states of each gene are: *w* (no mutation), *s* (somatic), and *b* (germline).

**Figure 6 ijms-24-09323-f006:**
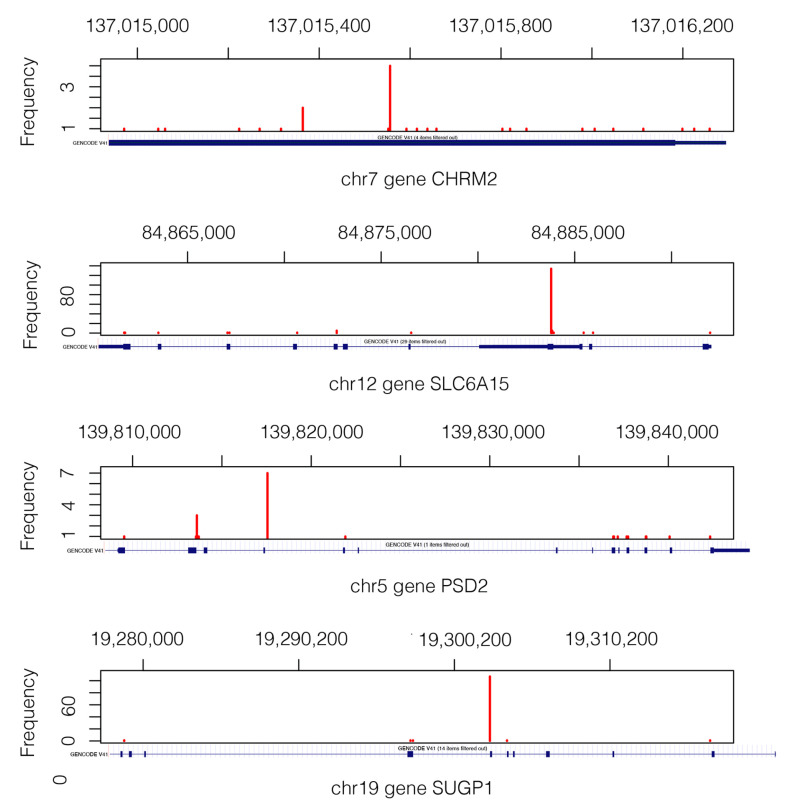
Mutation landscape in genes CHRM2, SLC6A15, PSD2, and SUGP1 from 405 subjects. The gene structure is shown below each gene figure. Mutations of gene CHRM2 are in the short range of 1.3 Kb, which is why the whole gene structure shown corresponds to an exonic region.

**Table 1 ijms-24-09323-t001:** Analysis of COAD samples. List of 16 gene pairs with a survival difference *p*-value lower than 0.05 sorted according to *p*-value for epistasis. The final columns indicate the hazard ratio (HR) and its corresponding confidence interval at a 95% level (HR 95% CI). SA *p*-value: Survival analysis *p*-value calculated comparing the overall survival rates of the groups of subjects with RGPMs and BSGMs. The contingency tables and survival curves for each gene pair in this table are reported in Appendix A and Appendix A of the Appendix A.

Gene1	Gene2	Epistasis *p*-Value	SA *p*-Value	HR	HR 95% CI
CCDC73	HTR2B	<1.04 ×$10^{-10}$	0.032	2.2	1.06–4.49
DDX4	KCNJ16	<1.04 ×$10^{-10}$	0.018	2.1	1.12–4.01
DDX4	SNX13	<1.04 ×$10^{-10}$	0.045	1.9	1.01–3.60
SEMG1	CYP2E1	<1.04 ×$10^{-10}$	0.033	1.6	1.04–2.43
TRIP12	BTAF1	$2.1\times 10^{-10}$	0.0042	2.7	1.34–5.45
ZNF99	HECTD2	$3.1\times 10^{-10}$	0.0065	2.1	1.21–3.54
LGR5	MBD5	$4.1\times 10^{-10}$	0.038	1.9	1.03–3.68
TOPORS	FLT1	$4.1\times 10^{-10}$	0.022	2.4	1.12–5.35
ABCA8	C1orf168	$6.2\times 10^{-10}$	0.044	1.5	1.01–2.22
MAGI3	KIF20A	$6.2\times 10^{-10}$	0.046	2.0	1.01–4.14
RASAL2	TRIM37	$6.2\times 10^{-10}$	0.014	2.8	1.20–6.55
PHLPP1	CLASP1	$6.2\times 10^{-10}$	0.021	2.2	1.11–4.46
ZNF491	BTAF1	$6.2\times 10^{-10}$	0.027	2.3	1.09–4.81
GTF3C3	CATSPERB	$7.3\times 10^{-10}$	0.011	1.9	1.15–3.12
MTNR1B	VIT	$7.3\times 10^{-10}$	0.046	1.7	1.01–2.71
ARHGAP20	ZBTB39	$7.3\times 10^{-10}$	0.036	2.0	1.04–3.82

**Table 2 ijms-24-09323-t002:** Contingency tables of the CCDC73-HTR2B (**A**) and DDX4-KCNJ16 (**B**) gene pairs. The number of patients holding relevant gene-pair mutations (RGPMs) correspond to the contingency table elements (s,b), (b,s), and (s,s). The mutation states of each gene are: *w* (no mutation), *s* (somatic), and *b* (germline).

(A) CCDC73-HTR2B					(B) DDX4-KCNJ16			
$g_{1}/g_{2}$	*w*	*s*	*b*		$g_{1}/g_{2}$	*w*	*s*	*b*
*w*	359	1	18		*w*	282	3	23
*s*	7	8	2		*s*	0	6	1
*b*	27	0	0		*b*	100	1	6

**Table 3 ijms-24-09323-t003:** Gene pairs for lung cancer that have a significant difference in survival with a *p*-value under 0.05 among the top 100 pairs according to the epistasis significance. The final columns indicate the hazard ratio (HR) and its corresponding confidence interval at a 95% level (HR 95% CI). SA *p*-value: Survival analysis *p*-value calculated comparing the overall survival rates of the groups of subjects with RGPMs and BSGMs. The contingency tables and survival curves for each gene pair in this table are reported in Appendix A and Appendix A of the Appendix A.

Gene1	Gene2	Epistasis *p*-Value	SA *p*-Value	HR	HR 95% CI
CHRM2	SLC6A15	$1.2\times 10^{-8}$	0.042	1.9	1.14–3.60
PSD2	SUGP1	$3.5\times 10^{-7}$	0.006	2.6	1.21–5.66
UBR1	ACAD9	$5.2\times 10^{-7}$	0.031	2.2	1.04–4.51

**Table 4 ijms-24-09323-t004:** Contingency tables of the CHRM2-SLC6A15 (**A**) and PSD2-SUGP1 (**B**) gene pairs. The relevant gene-pair mutations (RGPM) correspond to the contingency table elements (s,b), (b,s), and (s,s). The mutation states of each gene are: *w* (no mutation), *s* (somatic), and *b* (germline).

(A) CHRM2-SLC6A15					(B) PSD2-SUGP1			
$g_{1}/g_{2}$	*w*	*s*	*b*		$g_{1}/g_{2}$	*w*	*s*	*b*
*w*	267	0	114		*w*	300	1	80
*s*	6	5	4		*s*	1	3	4
*b*	7	0	2		*b*	13	0	3

## Data Availability

Gene pair data files for COAD (Appendix A) and LUAD (Appendix A).

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
