# Peer review of "Identification of Driver Epistatic Gene Pairs Combining Germline and Somatic Mutations in Cancer"

_ijms, 2023, doi:10.3390/ijms24119323_

Round 1

Reviewer 1 Report

The authors have developed a new algorithm to discover the drivers for cancer. Pairs of genes are taken into account. Overall, the study was well designed. There are several suggestions for the authors before publication.

1. In the title “A New Method for Discovering Drivers Made of Epistatic Gene Pairs in Cancer Tumors”, what is the meanings of cancer tumor? What is the difference of cancer and tumor in this algorithm?

2. In Figure 1, the picture seems nothing to do with the figure legend. The authors should further polish the legend.

3. What is carcinoma in Figure 1 and 4? Is it different from specific cancers or tumors in this algorithm?

4. The authors did not validate this new algorithm by making good use of clinical information in TCGA cancer cohorts except for survival data.

Reviewer 2 Report

In this paper authors propose a new method for the identification  of pairs of genes jointly mutated in tumor tissue not by chance. Such gene pairs simultaneous mutations are then checked to understand their role as a prognostic factors.

The topic is certainly interesting.

Major issues.

1. The x* is established to minimize the g-test statistics. The used p-value is then the one that comes from this optimization problem. Authors should better motivate and describe this choice because form the paper is not clear. If x is not minimized and the g-test statistics is computed directly from the data how are the results? Please clarify. Also the P-values shown in tables 1 and 2 are those coming from x*. Please clarify. Also, put within tables the Bonferroni FWER. In this case authors could apply instead FDR.  

2. From he top pairs of page 7 and 11 (which are not referenced trough the text) the values for o_ss are very low. Authors could try to use a pan-cancer approach and identify such pairs across the whole TCGA.

-In the survival analysis HR should be reported for all the pairs. Furthermore, how survival differences change if patients within o_sb or o_bs are put with those is o_ss?

Round 2

Reviewer 1 Report

I do not have further suggestions for the authors.

Reviewer 2 Report

The presentation of the results is still not rigorous. Important details are missing.

1. Survival curves are still not clear to the reviewer: 

-Which survival of TCGA has been used? OS? DFI? this has to be specified.

-HR CIs have to be added.

- A table on the bottom of each curve with the number of patients has to be added. 

2. Contingency tables of each pair together with survival curves have to be provided in supplementary materials.

Round 3

Reviewer 2 Report

Authors addressed all the reviewer's comments.